# Molecular characterization of plant growth-promoting vermi-bacteria associated with *Eisenia fetida* gastrointestinal tract

Saiqa Andleeb[1]*, Irsa Shafique[1], Anum Naseer[1], Wajid Arshad Abbasi[2], Samina Ejaz[3], Iram Liaqat[4], Shaukat Ali[4], Muhammad Fiaz Khan[5], Fayaz Ahmed[5], Nazish Mazhar Ali[4]

1 Department of Zoology, Microbial Biotechnology and Vermi-Technology Laboratory, University of Azad Jammu and Kashmir, Muzaffarabad, Pakistan, 2 Department of CS&IT, Computational Biology and Data Analysis Laboratory, University of Azad Jammu & Kashmir, Muzaffarabad, Pakistan, 3 Department of Biochemistry, Islamia University Bahawalpur, Bahawalpur, Pakistan, 4 Department of Zoology, Government College University, Lahore, Pakistan, 5 Department of Zoology, Hazara University, Mansehra, KPK, Pakistan

* drsaiqa@gmail.com, drsaiqa@ajku.edu.pk

**Data Availability Statement:** All relevant data are within the paper and at URLs provided in the Supporting information file.

## Abstract

Earthworms are highly productive invertebrates and play a vital role in organic farming and improving soil structure and function. The gastrointestinal tract of earthworms possessed agricultural important bacteria. So, the current research aimed was to examine, screen, and identify the plant growth promoting bacteria existing in the digestive tract of *Eisenia fetida* called plant growth promoting vermi-bacteria. The plant growth promoting traits such as siderophore, phytohormone, and hydrolytic enzymes production, and phosphate solubiliation were assessed. Eleven vermi-bacteria i.e. *Bacillus mycoides*, *B. aryabhattai*, *B. megaterium*, *Staphylococcus hominis*, *B. subtilis*, *B. spizizenii*, *B. licheniformis*, *B. mojavensis*, *B. toyonensis*, *B. anthracis*, *B. cereus*, *B. thuringiensis*, and *B. paranthracis* were isolated and identified based on microscopic studies, biochemical tests, ribotyping, and agricultural traits. All vermi-bacteria are Gram-positive rods except *Staphylococcus hominis* and produce different compounds such as siderophore, indole acetic acid, catalase, oxidase, proteases, amylases, and lipases. All vermi-bacteria also act as phosphate solubilizers. Therefore, all isolated vermi-bacteria could be used as potential microbial biofertilizers to enhance crops production in Pakistan.

## Introduction

Earthworms play a vital role in soil productivity, nutrient recycling, soil structure, and agriculture [1–3]. Thus, earthworms may be observed as a biological indicator of soil fertility and health [4, 5]. Guts of earthworms are suitable habitats for bacteria, and fungi, and proved that microbial numbers in the gut are much more compared to soil in which earthworms were living [6–8]. The gut environment is anoxic, with 6.9 pH having 50% water contents, enriched in total carbon, nitrogen, and organic carbon [9, 10]. Medina-Sauza et al. [11] showed that the growth of beneficial microbes in soil belongs to various families such as *Actinobacteria*,

**Funding:** Whole research work is done under HEC funded project No. TDF-02-006 titled: "Establishment of the vermi-tech unit at Azad Jammu and Kashmir, Muzaffarabad for vermi product development". The funders had no role in study design, data collection and analysis, decision to publish, or preparation of the manuscript.

**Competing interests:** The authors have declared that no competing interests exist.

*Proteobacteria*, *Firmicutes*, *Nitrospirae*, *Planctomycetes*, *Acidobacteria*, *Bacteroidetes*, and *Chloroflexi* are increased in number where earthworms are present. For each gram of vermi-compost, bacteria range from 102 to 106 [12]. Earthworms ingest PGPB such as *Rhizobium*, *Azotobacter*, *Bacillus*, *Azosprillium*, and *Pseudomonas* and increased in the gut up to 1000 fold due to the micro-environment of earthworm's gut [12, 13].

Santiago, [14] reported that seven different species of *Bacillus* have been identified from the digestive tract of *O. borincana*. By studying bacterial variety within the intestine of earthworms, Various methods and techniques were used for the identification of *Klebsiella*, *Bacillus*, *Azotobactor*, *Pseudomonas*, *Aeromonas*, *Serratia*, and *Enterobacter* [15, 16]. Sivasankari et al. [17] isolated 19 bacterial strains (*Escherichia* spp., *Micrococcus* spp, *Pseudomonas* spp., *Bacillus* spp., *Klebsiella* spp., *Erwinia* spp., *Streptococcus* spp., *Alcaligenes* spp. and *Enterobacter* spp.) from vermi-sources and screened for IAA (indole acetic acid) production. Six actinomycetes were isolated from 6 herbal vermi-composts and they produced PGP traits like siderophores, indole acetic acid, and enzymes (lipase, chitinase, and protease) [18]. Pandya et al. [19] investigated microbial variety (*Pseudomonas stutzeri* and *Pseudomonas mosselii*) from 3 vermi-compost samples produced in India. They inoculated 'MBCU1' and 'MBCU3' with the groundnut and chickpea plants and showed a rise in vegetative growth parameters compare to control which was un-inoculated.

Therefore, the purpose of the current research was to isolate and identify the vermi-bacteria from the gastrointestinal tract of *E. fetida* and to screen the agricultural traits of vermi-bacteria. These vermi-bacteria could be used in the field of agriculture and horticulture in Pakistan as a potential source of microbial biofertilizers compared to agrochemicals.

## Materials and methods

### Ethical statement

All experiments conducted during research work have been specifically designed to avoid any distress, suffering, and unnecessary pain to the experimental animals. All procedures were performed following international regulations referred to as Wet op de dierproeven (Article 9) of Dutch Law.

### Chemicals, glassware, equipment used

Nutrient broth (LENNOX), Nutrient agar medium (SIGMA-ALDRICH), Luria Bertani (LB) broth (LENNOX), McConkey agar (SIGMA-ALDRICH), mannitol salt agar (OXOID), skim milk agar (NEOGEN), 3% KOH, starch (SIGMA-ALDRICH), Gram staining kit (MERCK), bacteriological peptone (OXOID), hydrogen peroxide, Kings B medium (SIGMA), Wattman No. 1 disc, oxidase reagent, phenol, 0.5% picric acid (SIGMA-ALDRICH), Kovacs reagent, 2% Sodium carbonate (MERCK), Nessler's reagent (SIGMA-ALDRICH), dilute iodine, Lead (III) nitrate (Sigma- Aldrich), cadmium nitrate tetrahydrate (Sigma- Aldrich), chromium (III) nitrate (Sigma- Aldrich). Analytical balance (SARTORIUS GMBM GOTTINGEN, Germany), digital weighing machine (Jeweler Precision Balance Model: DH-V600A)steam sterilizer (autoclave), 37ºC incubator (MMM group Medcenter Enrich tungsten GmbH), 37ºC shaker (Irmeco GmbH, Germany), Laminar flow (ESCO Prod Model; EQU/03-EHC; Serial # 2000–0052), sterile dissecting pins, Sterile distilled water, dissecting box, gloves, dissecting board, sterile bottles, 70% ethanol, 500 ml beakers, micropipette, 250 ml conical flasks, test tubes, bacteriological wire loop, Petri plates, glycerol, glass rod, glass slides, coverslips, spirit lamp, microscope, and toothpicks.

## Sampling and dissection of *E. fetida*

To isolate vermi-bacteria, 4–5 mature clitellate *E. fetida* were collected and taken to the vermi-technology laboratory, Zoology Department, University of AJ&K (UAJ&K), Muzaffarabad. Mature earthworms were rinsed with sterile distilled water, cleaned externally with 75% ethanol in a sterile Petri plate, and dried with tissue papers. With its anterior end pointing forward, they were placed around the second, third, and fourth fingers of the left hand. Sterilized pair of dissecting scissors were used with their fine sharp tip introduced into the ventral region at the clitellum, with the help of scissors body wall was slightly raised and the cut was gently made along the length of the worm. Earthworm was held down on a board, with the help of sterile dissecting pins, and the body wall was stretched to expose the internal organs. With sterilized forceps, the gut and nephridia were then freed from surrounding blood vessels and separated into foregut, midgut, and hindgut. After washing the parts of the gut with sterile distilled water, they were suspended in another sterilized bottle containing distilled water (10 ml). The homogenized mixture was used further for bacteria isolation.

## Isolation and enumeration of vermi-bacteria

Somasegaran and Hoben's [20] used the serial dilution method for the isolation of bacteria. A homogenized mixture (1 ml) was poured into another test tube to make $10^{-1}$ dilution. Similarly, other dilutions $10^{-2}$, $10^{-3}$, $10^{-4}$, $10^{-5,}$ and $10^{-6}$ were prepared accordingly. After making dilutions, the mixture was spread on a nutrient agar medium and placed for 24 h at 37°C. The diverse bacterial colonies developed on the media were estimated and expressed as colony-forming units (CFU). The concentration of bacteria in the original sample was calculated as:

$$Colony\ forming\ unit(CFU) = \frac{Number\ of\ Colonies X Diluation\ factor}{Volume\ of\ Inoculum}$$

## Bacterial purification

From the well-separated dilution plates, a total of eleven bacterial strains were picked, grown in a nutrient broth medium, and placed for 24 h at 37°C. The next day, overnight culture was streaked on freshly prepared nutrient agar plates and incubated at 37 °C for 24 h. These plates were labeled as U1, U2, U3, U4, U5, U6, B1, B2, B3, B4, and B6, respectively. After sub-culturing, these eleven vermi-bacterial isolates were picked and stored in 60% glycerol for future work.

## Morphological and biochemical characterization of vermi-bacteria

Gram staining and different media (MacConkey and nutrient agar) were used to study the morphological features of vermi-bacteria. A loop full of glycerol stock was dipped into a nutrient broth medium and incubated at 37 °C for 24 h. After incubation, overnight culture was spread on MacConkey agar and nutrient agar plates and incubated for 24 h at 37 °C. After incubation, the colony characteristics such as motility, colony shape, color, the shape of the cell, elevation, margin, and texture were recorded. All vermi-bacteria were screened for biochemical tests (catalase, oxidase, urease, citrate, lipolytic, proteolytic, amylolytic, and mannitol fermentation) and plant growth-promoting traits (Potassium hydroxide test, Hydrogen cyanide production, Indole acetic acid production, Ammonia production test, Phosphate solubilization, Siderophore production tests [21–35].

## Genomic DNA extraction

From vermi-bacterial isolates, genomic DNA extraction was carried out using the method of Sambrook et al. [36], with slight modifications. All vermi-bacterial isolates were grown in a Luria broth medium and incubated for 24 h at 37 ˚C. After incubation, centrifugation was carried out at 10,000 rpm for 5 min to harvest cells. The pellet was suspended in lysis buffer-1 (Tris EDTA and SDS; pH 4.0) and then centrifuged for 10 min at 10,000 rpm. After centrifugation, 500 $\mu$L of chloroform: isoamyl alcohol (24: 1) was added, mixed, and centrifuged at 10,000 rpm for 10 min. In the collected supernatant 2.5 volumes of chilled 100% absolute ethanol and $1/10^{th}$ volume of sodium acetate was added and incubated at -20 ˚C overnight. The next day, samples were centrifuged for 10 min at 10,000 rpm, and the pellet was washed with 70% ethanol. After centrifugation, pellets were dried for 3 h, and DNA was dissolved in distilled water (20 $\mu$l).

## Amplification and sequencing of 16S rDNA

For the identification of vermi-bacterial isolates, full-length 16S rRNA primers (341F 5′ – CCTACGGGNGGCWGCAG–3′; 806R 5′ –GGACTACNNGGGTATCTAAT–3′) were taken to amplify the V3-V4 region (Approx. 470 bps) using following PCR conditions (initial denaturation 95˚C for 2 min; Cyclic denaturation at 95˚C for 20 sec; Annealing at 50˚C for 30 sec; Cyclic extension 72˚C for l min 30 sec; and Final extension 72˚C for 5 min; 35 cycles). After PCR analysis, all PCR products were sent to Macrogen, Korea for sequence analysis. The obtained nucleotide sequences further proceeded for homology through BLAST at National Center for Biotechnology Information (NCBI) platform.

## Phylogenetic analysis

The phylogenetic relationship was determined using the Maximum Likelihood method and Tamura-Nei model [37]. This analysis involved 32 nucleotide sequences. There were a total of 1573 positions in the final dataset. Evolutionary analyses were conducted in MEGA X [38]. After BLAST and phylogenetic analysis, all amplified sequences were submitted to NCBI, Genbank for the provision of accession numbers.

# Results

## Morphological and biochemical characterization of vermi-bacteria

Several well-separated colonies were observed in the case of $10^{-3}$, $10^{-4}$, and $10^{-5}$ diluted samples. These plates were selected for the isolation of pure vermi-bacterial isolate. Eleven bacterial strains were isolated from these dilutions such as 2 isolates from $10^{-3}$, 3 isolates from $10^{-4}$, and 6 isolates from $10^{-5}$, respectively. These vermi-bacterial isolates were further grown on the nutrient agar, mannitol salt agar, and MacConkey agar, respectively. Out of 11 isolates, U1, U2, and U3 form yellow colonies with yellow zones indicating their ability to ferment mannitol whereas U4, U5, U6, B1, B2, B3, B4, and B6 showed no growth in mannitol salt agar. Out of 11 isolates U2, U4, U5, B3, and B6 produced pink colonies indicating their ability to ferment lactose. Whereas U1, U3, U6, B1, B2, and B4 form yellow zones indicating that cannot ferment the lactose (Fig 1). The color of colonies was creamy, yellow, and white, showing entire and undulate margins, the elevation of colonies was also varied from raised flat, and convex, and colonies' forms were circular, punctiform, and irregular, respectively. Most colonies were translucent and opaque. The shape of vermi-bacterial isolates was also varied from singly rods coccobacilli, and branching rods. All isolated vermi-bacteria were Gram-positive rods (Fig 2; Table 1).

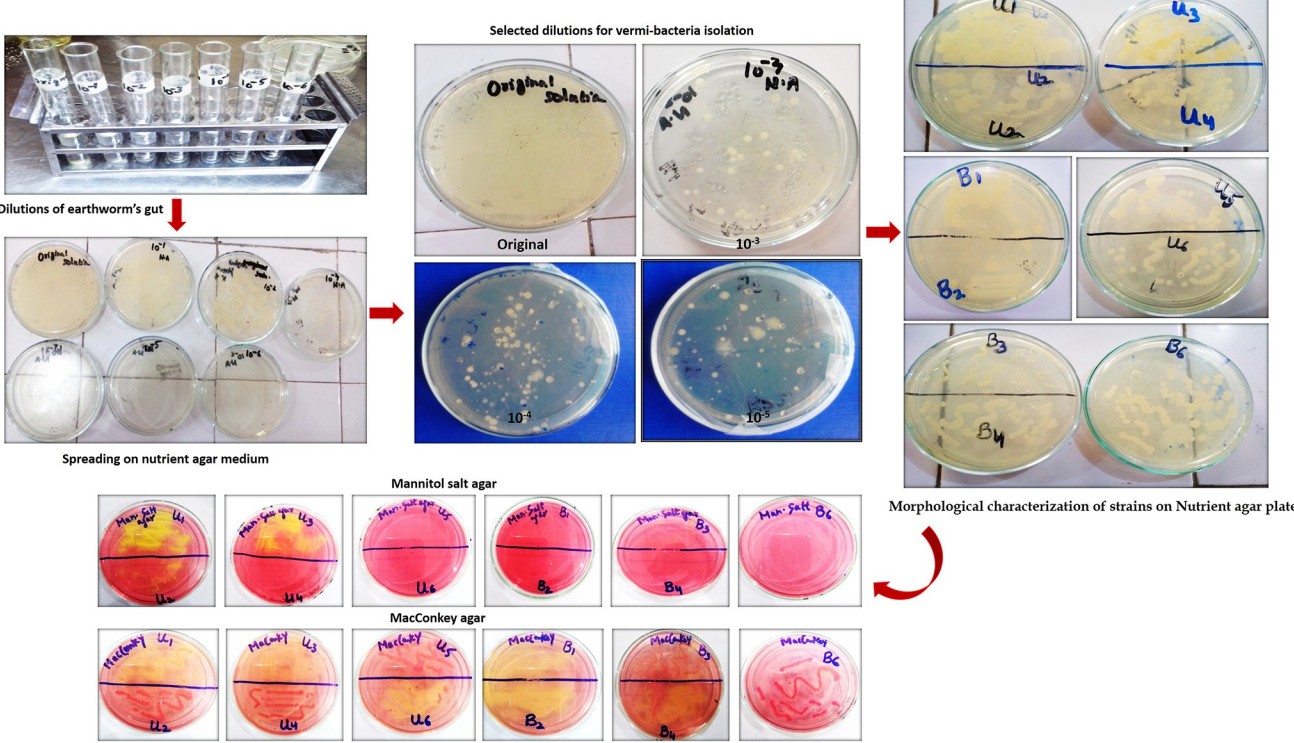

**Fig 1. Isolation and screening of vermi-bacterial isolates using different culturing media.**

## Biochemical characterization

The results of the KOH test showed that all isolates were Gram-positive bacteria, and their cell wall was not affected by 3% KOH and do not form stringy, sticky, and viscous material within the first 30 sec (Table 1). All vermi-bacterial isolates form a cherry red ring at the top of the medium denoting the indole production action (Fig 3). All vermi-bacteria did not show yellow color which means that all vermi-bacteria cannot produce ammonia (Table 1). All vermi-bacterial isolates were not able to produce HCN as the color of the filter paper did not change (Table 1). All vermi-bacterial isolates were catalase-positive. Similarly, all vermi-bacterial isolates were oxidase-positive except U2 and B3 as purple color spots appeared within 5–10 secs on the filter paper after adding the oxidase reagent (Table 1; Fig 3). All vermi-bacterial isolates showed amylase and lipase production (Table 1; Fig 3). All vermi-bacterial showed proteolytic activity in the range of 10.0 ± 0.0 mm to 15.0 ± 0.0 mm except B1 and B2 (Fig 3). The maximum clear zone for lipolytic activity was recorded in the range of 15.0 ± 0.0 mm to 20.0 ± 0.0 mm (Fig 3). All bacterial isolates were screened for siderophore production on CAS agar plates and results revealed that all isolates showed siderophore production. All vermi-bacterial isolates produced maximum siderophore except U3, B3, and B6. The activity diameter was recorded in the range of 12.0 ± 0.0 mm to 33.0 ± 0.0 mm. The zone of activity indicated the amount of siderophore excreted by the bacterial isolates (Fig 3). Results revealed that all bacterial isolates were phosphate solubilizers. The clear zone around the colonies indicated positive phosphate solubilization activity. The clear zones were recorded to have a range of 12.0 ± 0.0 mm to 33.0 ± 0.0 mm (Fig 3).

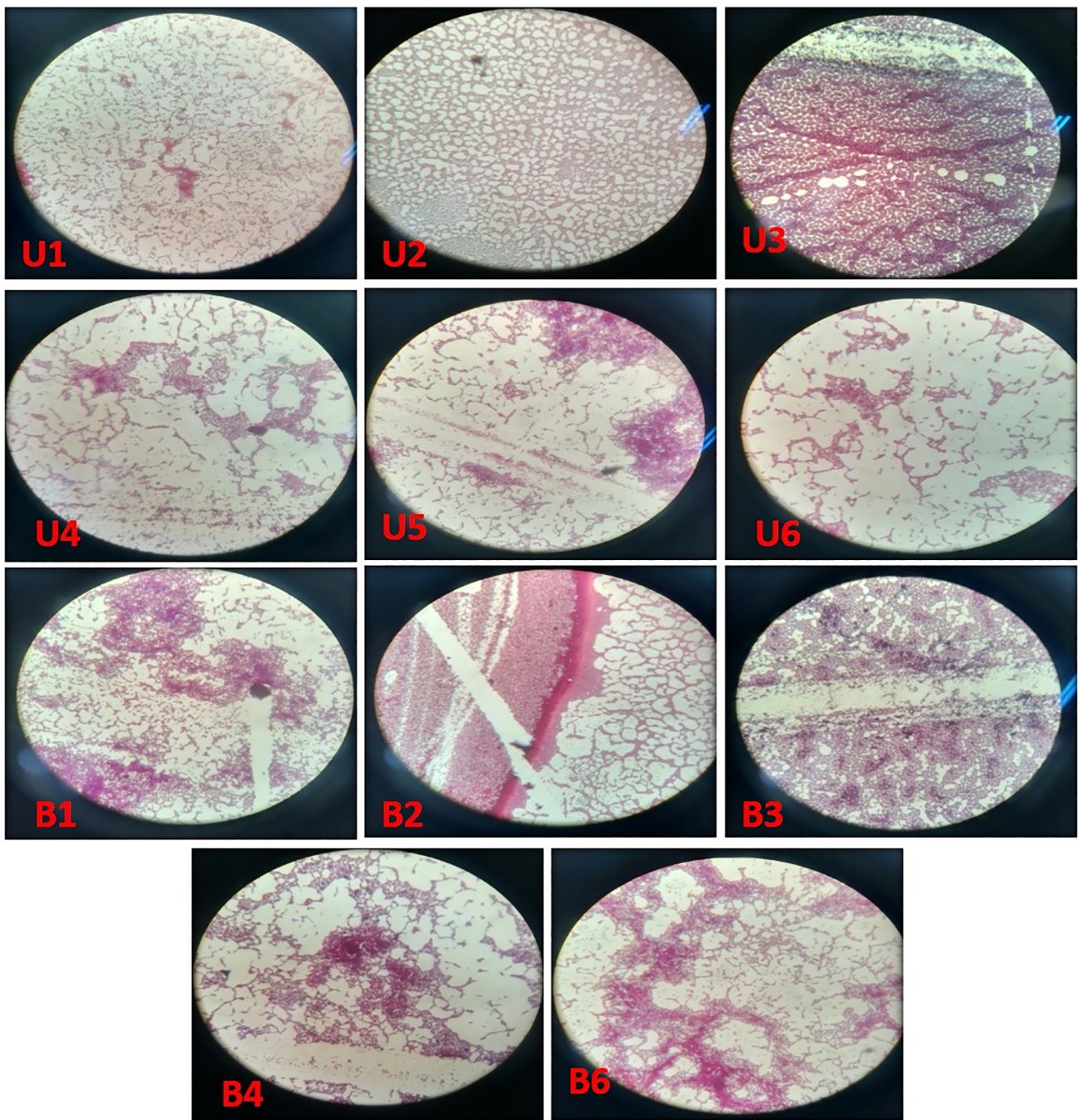

**Fig 2. Gram's staining of vermi-bacterial isolates.**

## Molecular characterization

After morphological and biochemical characterization, genomic DNA was extracted, 16S rRNA was amplified, and sequenced. The range of amplified PCR products was 419 bps-443 bps and the length of obtained accessions (419 bps-1517 bps) is shown in Table 2. The results indicated 94.94% to 100% homology with various bacterial species such as B1 showed 100%

**Table 1. Morphological and biochemical characterization of plant growth promoting vermi-bacteria associated with *E. fetida*.**

| Characteristics → / Bacterial strains ID↓ | GS | Sha | Color | Colony shape | Ele | Mar | Tex | Mot | Sid | Phos | KOH | IAA | HCN | NH₃ | Cata | Oxid | Amy | Pro | Lip | Ure | Cit |
|---|---|---|---|---|---|---|---|---|---|---|---|---|---|---|---|---|---|---|---|---|---|
| *Bacillus mycoides (B1, B4 U3)* | + | Rod | White | Spiral/circular | Conv | Ent | Wet | - | + | + | - | - | - | + | + | - | + | + | + | - | - |
| *Bacillus/Priestia megaterium (B2)* | + | Rod | Cream | Cir | Con | Ent | Wet | + | + | + | + | - | - | + | + | - | + | + | + | - | + |
| *Staphylococcus hominis (B3)* | + | cocci | Cream | Punc/irr | Flat | Ser | dry | + | + | + | - | + | - | - | + | - | + | + | + | - | + |
| *Bacillus subtilis (U1)* | + | Rod | White | Circular | Flat | Ent | dry | + | + | + | + | + | - | + | + | + | + | + | + | - | + |
| *Bacillus toyonensis (U4)* | + | Rod | White | Circular | Flat | Ent | Wet | + | + | + | - | + | - | + | + | + | + | + | + | - | + |
| *Bacillus thuringiensis (U5)* | + | Rod | White | Circular | Con | Ent | Wet | + | + | + | - | + | - | + | + | + | + | + | + | - | + |
| *Bacillus paranthracis (U6)* | + | Rod | White | Punct | Con | Ent | Wet | + | + | + | - | + | - | + | + | + | + | + | + | - | + |
| *Bacillus licheniformis (B6)* | + | Rod | White | Spiral/circular | Conv | Ent | Wet | - | + | + | - | +- | - | + | + | + | + | + | + | - | + |
| *Bacillus mojavensis (U2)* | + | Rod | Opa | Irr | Con | Ent | Wet | + | + | + | + | + | + | + | + | + | + | + | + | - | + |

Elevation (Ele), margin (Mar), texture (Tex), motality (Mot), siderophore (Sid), phosphate (Phos), potassium hydroxide (KOH), IAA (indole acetic acid), Hydrogen cyanide (HCN), ammonia (NH3), catalase (Cata), oxidase (Oxid), amylase (Amy), proteases (Pro), lipase (Lip), urease (Ure), citrate (Cit), Filamentous (Fil), Filliform (Filli), irregular (Irr), punctinate (Punc), convex (Con), flat, raised (Rai), umbonate (Umb), entire (Ent), serrate (Ser), dry, wet, and shine (Shi), + (positive),—(negative)

homology with *Bacillus anthracis* (MG733605.1), *B. thuringiensis* (MG208031.1), *B. cereus* (MH732105.1), *B. mycoides* (MN416959.1), *B. tyonensis* (MK038983.1). Similarly, other vermi-bacteria showed homology as: B2 showed 100% with *B. aryabhattai* (MF527247.1) and 100% with *B. megaterium* (KP893549.1); B3 showed 99.52% with *Staphylococcus hominis* (KM392087.1) and 100% with *Staphylococcus epidermidis* (KJ806213.1); B4 indicated 100% with *B. anthracis* (GQ392044.1), 100% with *B. thuringiensis* (MT510408.1), 100% with *B. cereus* (MT510411.1), 100% with *B. mycoides* (MN416959.1), 99.76% with *B. toyonensis* (MK038983.1), 99.76% with *B. amyloliquefaciens* (KY009547.1); B6 showed 99.29% with *B. cabrialesii* (MZ342760.1), 99.05% with *B. tequilensis* (MK611555.1), 98.81% with *B. velezensis* (MZ082985.1), 98.81% with *B. spizizenii* (MZ081559.1); U1 showed 99.29% with *B. spizizenii* (MZ317416.1), 99.29% with *B. cabrialesii* (MZ342760.1), 99.53% with *B. tequilensis* (MK611555.1), 99.29% with *B. licheniformis* (MZ331398.1), 99.53% with *B. subtilis* (MT273659.1); U2 showed 99.29% with *B. mojavensis* (MW659923.1), 99.06% with *B. flexus* (KT265075.1), 99.30% with *B. halotolerans* (KY127379.1), 99.06% with *B. xiamenensis* (MW843010.1), 99.28% with *B. subtilis* (MN726675.1); U3 showed 99.76% with *B. toyonensis* (MK038983.1), 99.76% with *B. pseudomycoides* (KY344825.1), 99.76% with *B. gaemokensis* (KY344805.1), 99.76% with *B. amyloliquefaciens* (KY009547.1), 99.76% with *B. paramycoides* (MT576619.1), 100% with *B. mycoides* (MN416959.1); U4 showed 99.76% with *B. cereus* (MT510411.1), 99.76% with *B. toyonensis* (MK038983.1), 99.76% with *Bacillus thuringiensis* (MT510408.1); U5 showed 99.76% with *B. gaemokensis* (KY344805.1), 99.76% with *B. paramycoides* (MT576619.1), 100% with *B. anthracis* (GQ392044.1), 99.76% with *B. amyloliquefaciens* (KY009547.1), 99.76% with *B. toyonensis* (MK038983.1); and U6 showed with 95.42% with *B. paranthracis* (MK547279.1), 94.94% with *B. wiedmannii* (MG726003.1), 94.94% with *B. tequilensis* (JX898005.1), respectively.

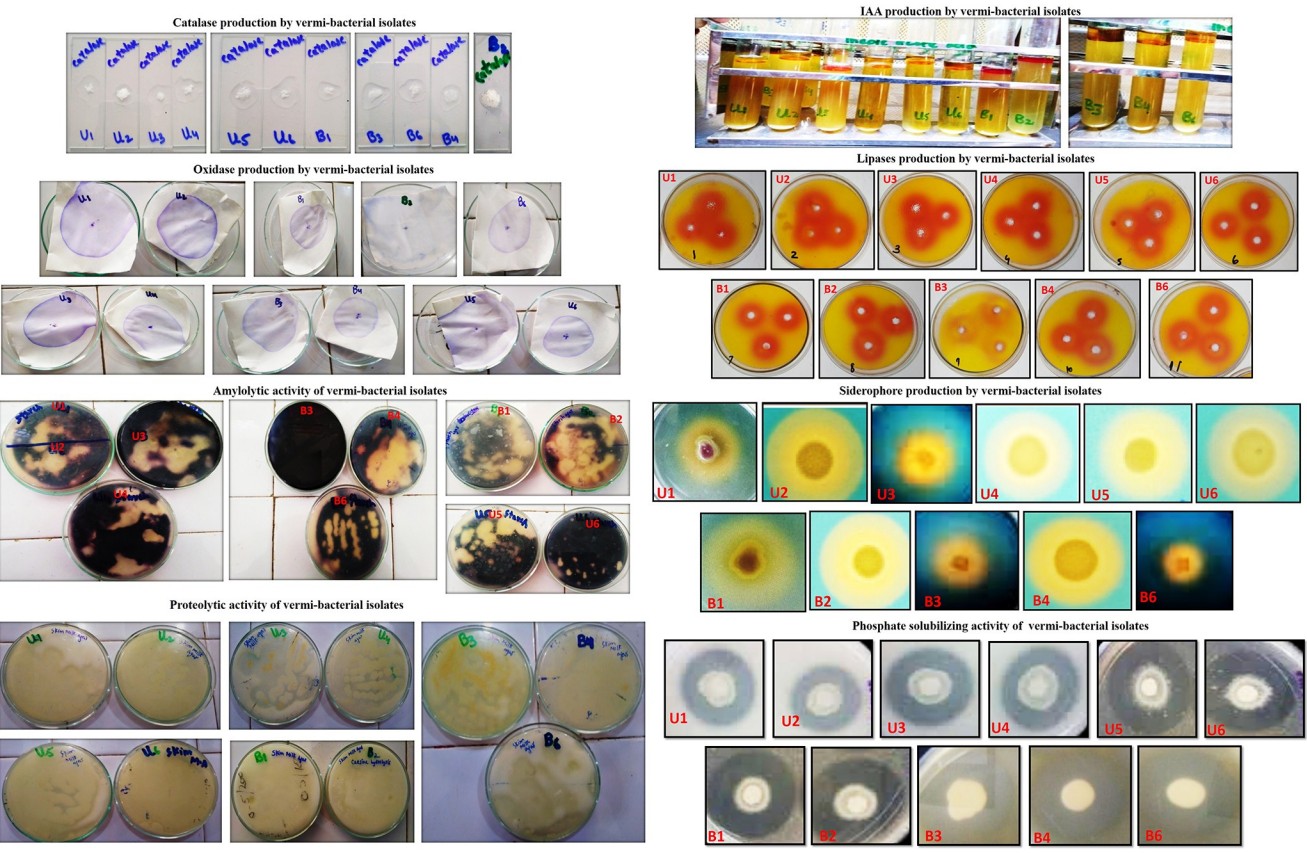

**Fig 3. Biochemical tests and plant growth promoting traits of vermi-bacterial isolates.**

**Table 2. Homology prediction of plant growth promoting vermi-bacteria using BLAST, NCBI genbank.**

| Code | Amplified PCR product size (bps) | Scientific name | Percentage identity | Accession Length (bps) | Accession |
|------|----------------------------------|-----------------|---------------------|------------------------|-----------|
| B1 | 413 | *Bacillus mycoides* | 100% | 653 | MN416959.1 |
| B2 | 440 | *Bacillus aryabhattai* | 100% | 425 | MF527247.1 |
| | | *Bacillus megaterium* | 100% | 419 | KP893549.1 |
| B3 | 442 | *Staphylococcus hominis* | 99.52% | 937 | KM392087.1 |
| B4 | 438 | *Bacillus mycoides* | 100% | 653 | MN416959.1 |
| B6 | 440 | *Bacillus subtilis* | 99.28% | 445 | MN726675.1 |
| | | *Bacillus spizizenii* | 98.81% | 1379 | MZ081559.1 |
| | | *Bacillus licheniformis* | 99.29% | 1116 | MZ331398.1 |
| U1 | 443 | *Bacillus subtilis* | 99.53% | 1169 | MT273659.1 |
| U2 | 443 | *Bacillus mojavensis* | 99.29% | 790 | MW659923.1 |
| U3 | 441 | *Bacillus mycoides* | 100% | 653 | MN416959.1 |
| U4 | 443 | *Bacillus toyonensis* | 99.76% | 550 | MK038983.1 |
| | | *Bacillus mycoides* | 100% | 653 | MN416959.1 |
| U5 | 438 | *Bacillus anthracis* | 100% | 562 | GQ392044.1 |
| | | *Bacillus cereus* | 99.76% | 591 | MT510411.1 |
| | | *Bacillus thuringiensis* | 100% | 591 | MT510408.1 |
| U6 | 436 | *Bacillus paranthracis* | 95.42% | 1412 | MK547279.1 |

On the other hand, the phylogenetic tree was constructed among the 16S rRNA obtained sequences of all vermi-bacterial isolates along with BLAST nucleotide sequences and results revealed that all vermi-bacterial isolates showed resemblance to the BLAST analysis (Figs 4–14; Table 2). The phylogenetic relationship among vermi-bacterial isolates and BLAST sequences was constructed using the Maximum Likelihood method and Tamura-Nei model (Figs 4–14). The vermi-bacterial isolate B1 was the most closely related to the *Bacillus mycoides* (MN416959.1) with 100% similarity in 16S rRNA sequences (Fig 4). Similarly, vermi-bacteria (B2), showed a cluster of two closely related species i.e. *Bacillus aryabhattai* (MF527247.1) and *Bacillus megaterium* (KP893549.1) (Fig 5), B3 showed resemblance *Staphylococcus hominis* (KM392087.1) and *Staphylococcus epidermidis* (KJ806213.1) (Fig 6), B4 closely related to *Bacillus mycoides* (MN416959.1) (Fig 7). The phylogeny cluster of Vermi-bacteria (B6) showed relation with *Bacillus subtilis* (MN726675.1), *Bacillus spizizenii* (MZ081559.1), *Bacillus licheniformis* (MZ331398.1), *Bacillus tequilensis* (MK611555.1), *Bacillus mojavensis* (MW659923.1), and *Bacillus flexus* (KT265075.1) (Fig 8). The cluster of U1 was closely related to *Bacillus subtilis* (MT273659.1), *Bacillus spizizenii* (MZ081559.1), *Bacillus licheniformis* (MZ331398.1), and *Bacillus flexus* (KT265075.1) (Fig 9), U2 closely related to 99.29% with *Bacillus mojavensis* (MW659923.1) (Fig 10), U3 closely related to *Bacillus mycoides* (MN416959.1) (Fig 11), U4 closely related to *Bacillus toyonensis* (MK038983.1) and *Bacillus mycoides* (MN416959.1) (Fig 12), U5 closely related to *Bacillus anthracis* (GQ392044.1) *Bacillus cereus* (MT510411.1), and *Bacillus thuringiensis* (MT510408.1) (Fig 13), and U6 closely related to *Bacillus paranthracis* (MK547279.1) (Fig 14), supporting the 100% value from bootstrap analysis of the phylogenetic trees. The scale bar in all figures represents 0.05% sequence divergence. The amplified vermi-bacteria sequences were submitted to Genbank and the provided accession numbers are given as *Staphylococcus hominis* (OL364179), *Bacillus mycoides* (OL364177), *Bacillus mycoides* (OL364180), *Bacillus mycoides* (OL364184), *Bacillus licheniformis* (OL364181), *Bacillus paranthracis* (OL364187), *Bacillus subtilis* (OL364182), *Bacillus megaterium/Priestia megaterium* (OL364178), *Bacillus toyonensis* (OL364185), *Bacillus thuringiensis* (OL364186), *Bacillus mojavensis* (OL364183), respectively.

## Discussion

The gut of earthworms consists of mucous (organic matters, proteins, and polysaccharides) and microbes (bacteria, fungi and protozoans). Earthworms gut provide a suitable environment to the microbes [13]. Khyade, [6], Byzov et al. [15], and Singleton et al. [16] reported the presence of *Klebsiella*, *Bacillus*, *Azotobacter*, *Pseudomonas*, *Aeromonas*, *Serratia*, and *Enterobacter* in the intestine of earthworms. Vijayakumar et al. [39] identified beneficial bacteria like *Pseudomonas sp*, *Bacillus sp*, *Cellulomonas sp*, *Micrococcus sp*, and *Escherichia coli* from the gut of *Perionyx excavates* using biochemical tests. In the current research, eleven vermi-bacteria such as *Staphylococcus hominis* (OL364179), *Bacillus mycoides* (OL364177), *Bacillus mycoides* (OL364180), *Bacillus mycoides* (OL364184), *Bacillus licheniformis* (OL364181), *Bacillus paranthracis* (OL364187), *Bacillus subtilis* (OL364182), *Bacillus megaterium/Priestia megaterium* (OL364178), *Bacillus toyonensis* (OL364185), *Bacillus thuringiensis* (OL364186), *Bacillus mojavensis* (OL364183) from the gut of *Eisenia fetida* were identified based on staining, morphological characteristics, biochemical tests, and 16SrRNA. Our findings agreed with Hyun-Jung et al. [40], who revealed that the Bacillus species are dominant in the intestine of earthworms. Based on the previous findings we can say that Bacillus species could be an active member of host microbiota.

Morphologically, colonies indicated the variations in the margin, elevation, color, form, opacity, and shape, respectively. Gram staining technique indicated that all isolated bacterial

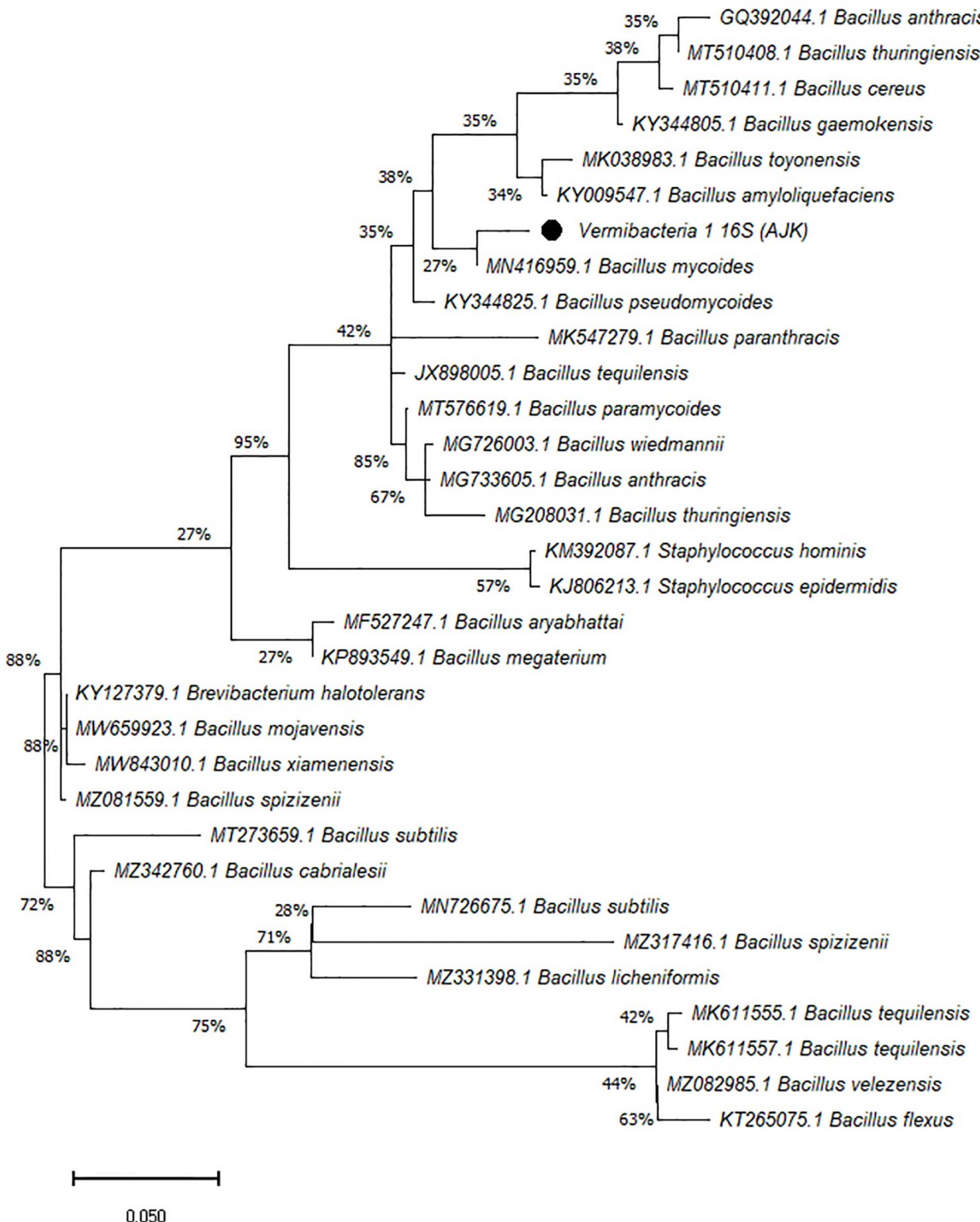

**Fig 4. Phylogenetic relationship of *Bacillus mycoides* with other known bacterial species.**

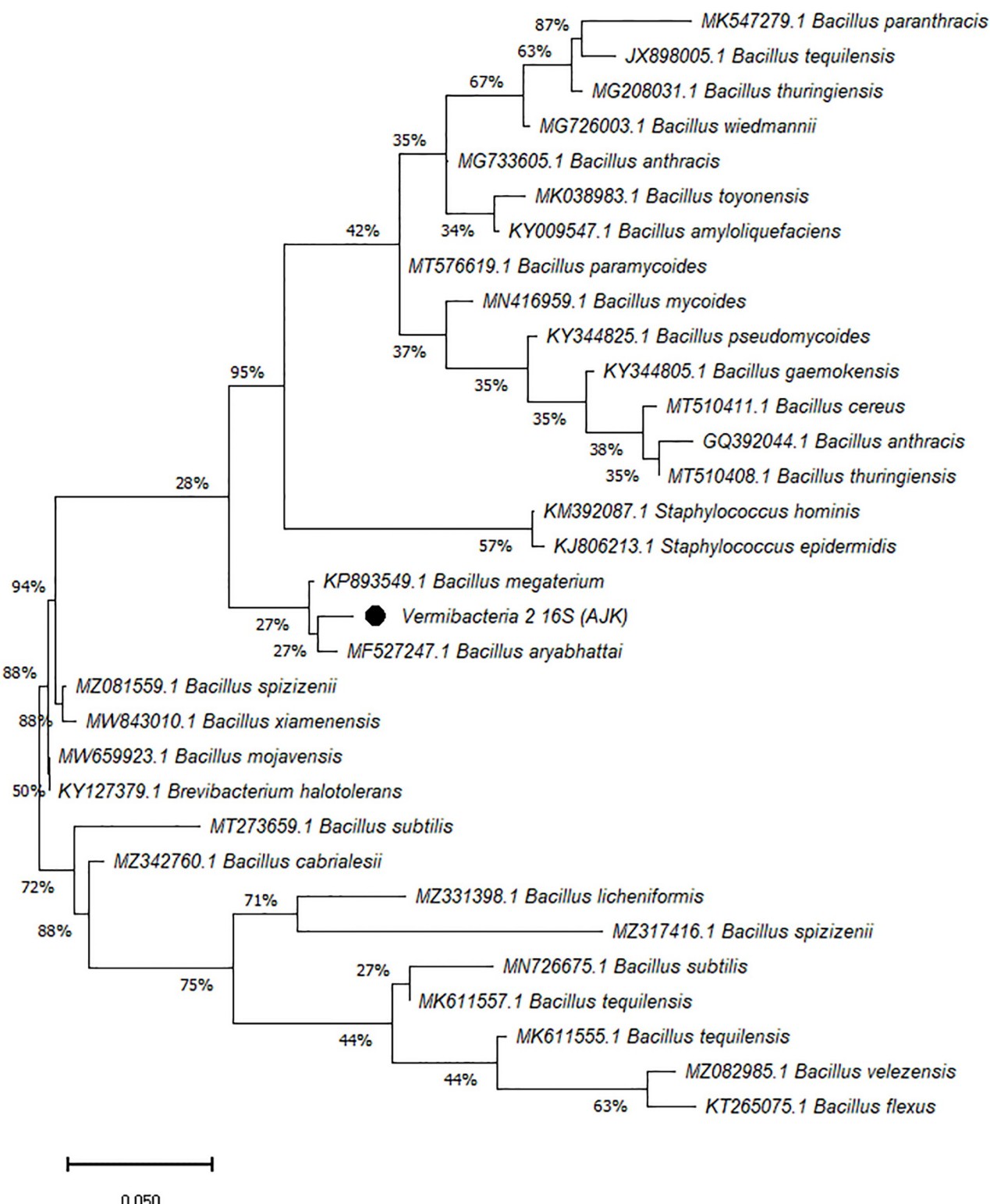

**Fig 5. Phylogenetic relationship of *Bacillus megaterium* with other known bacterial species.**

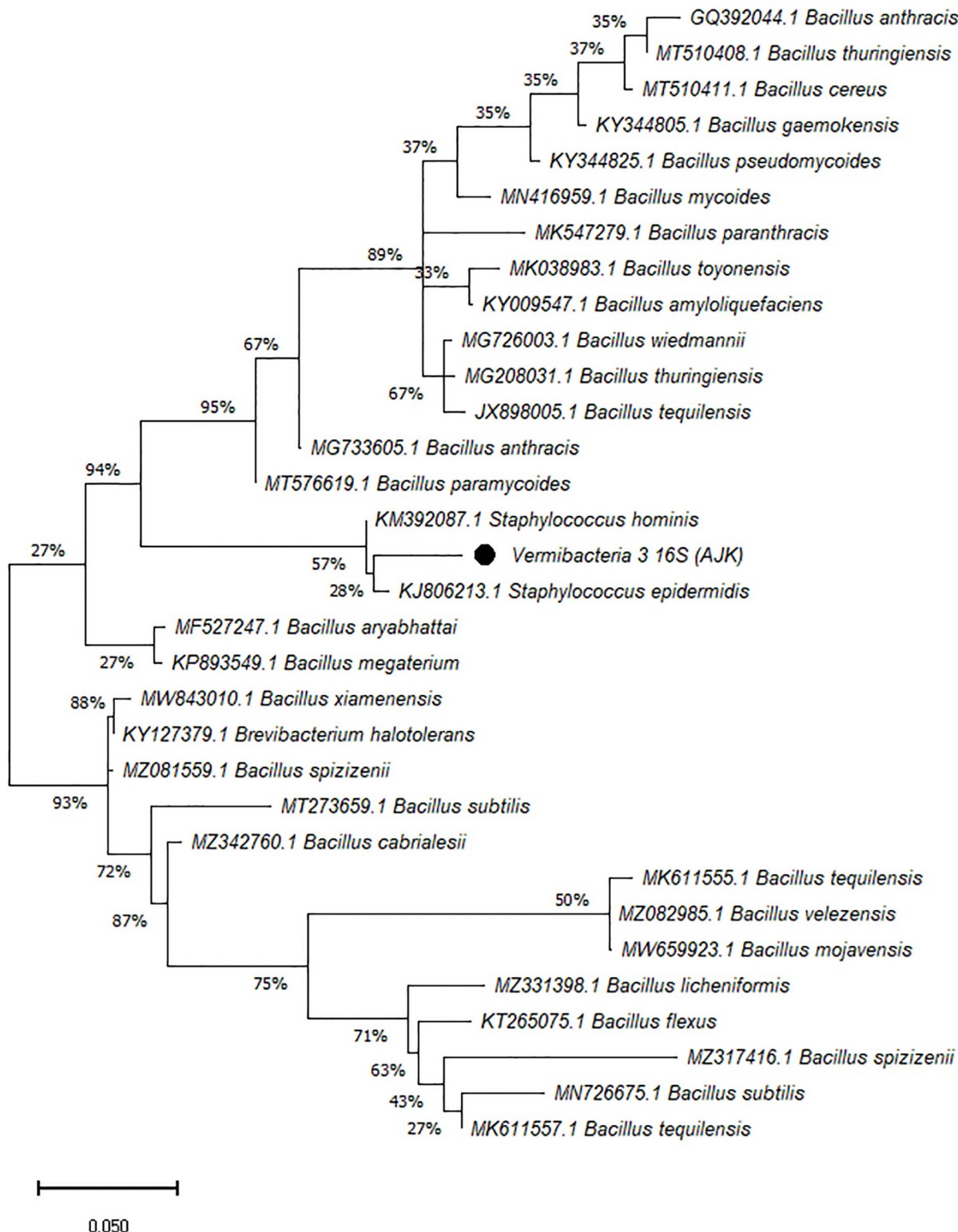

**Fig 6. Phylogenetic relationship of *Staphylococcus hominis* with other known bacterial species.**

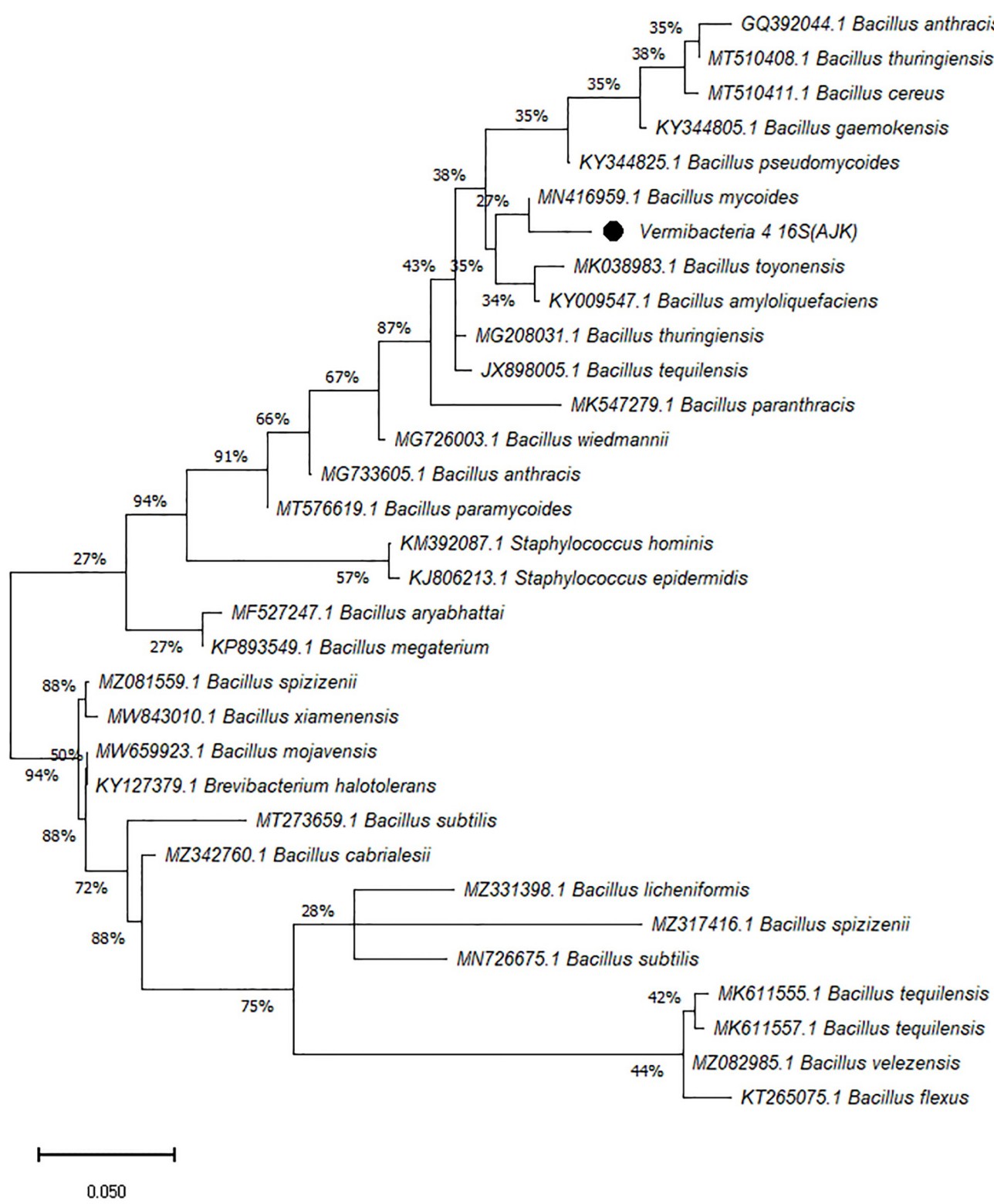

**Fig 7. Phylogenetic relationship of *Bacillus mycoides* with other known bacterial species.**

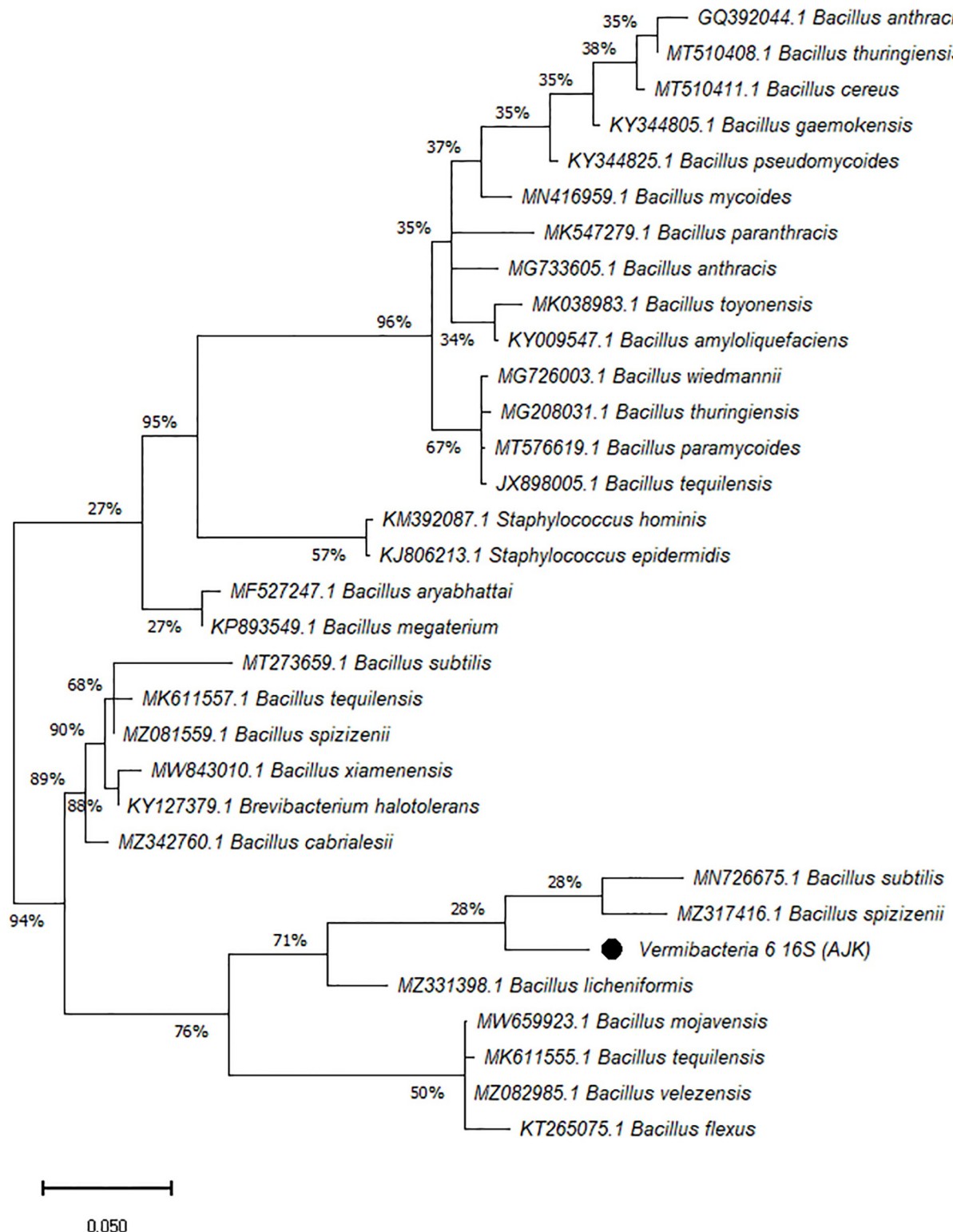

**Fig 8. Phylogenetic relationship of *Bacillus licheniformis* with other known bacterial species.**

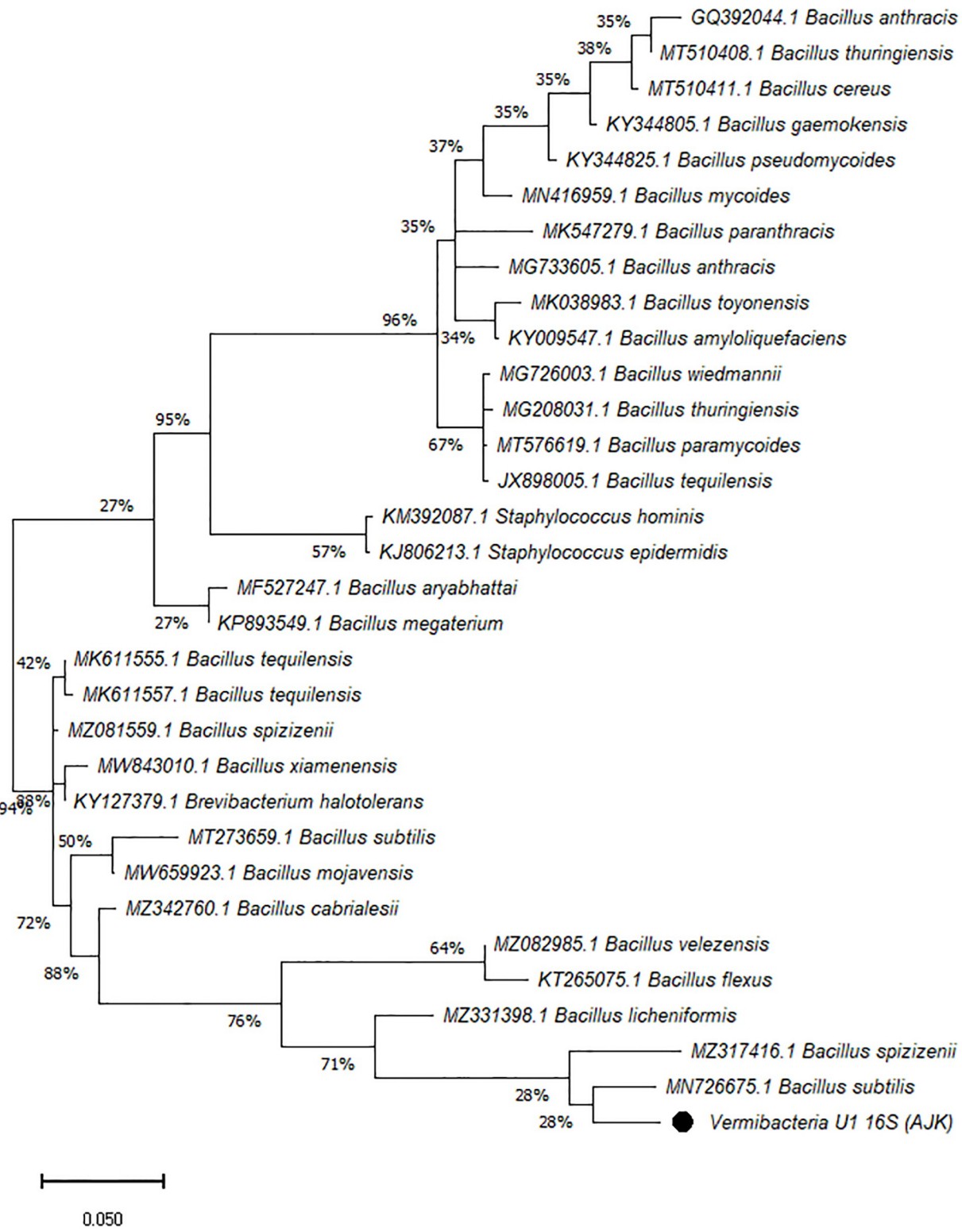

**Fig 9. Phylogenetic relationship of *Bacillus subtilis* with other known bacterial species.**

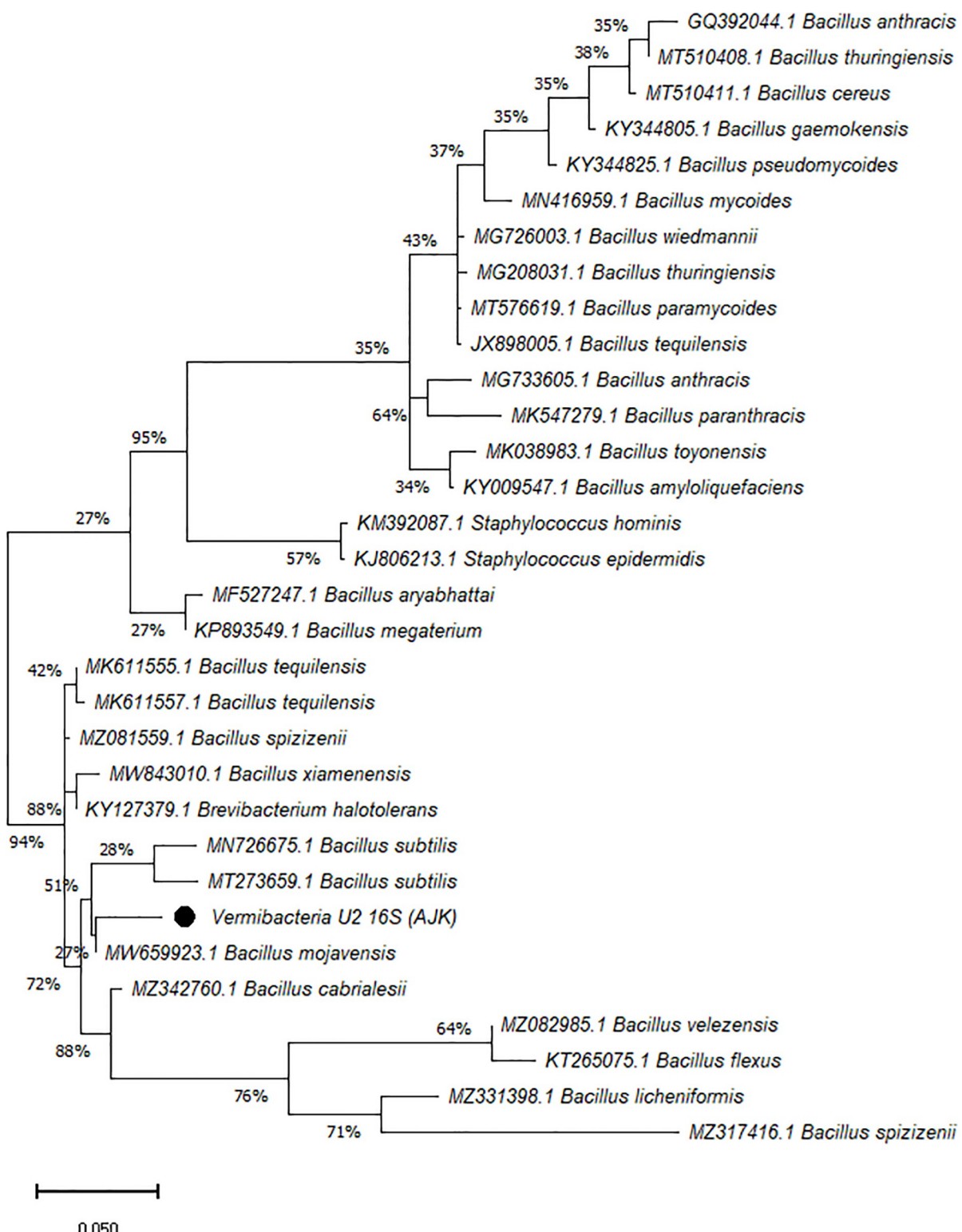

**Fig 10. Phylogenetic relationship of *Bacillus mojavensis* with other known bacterial species.**

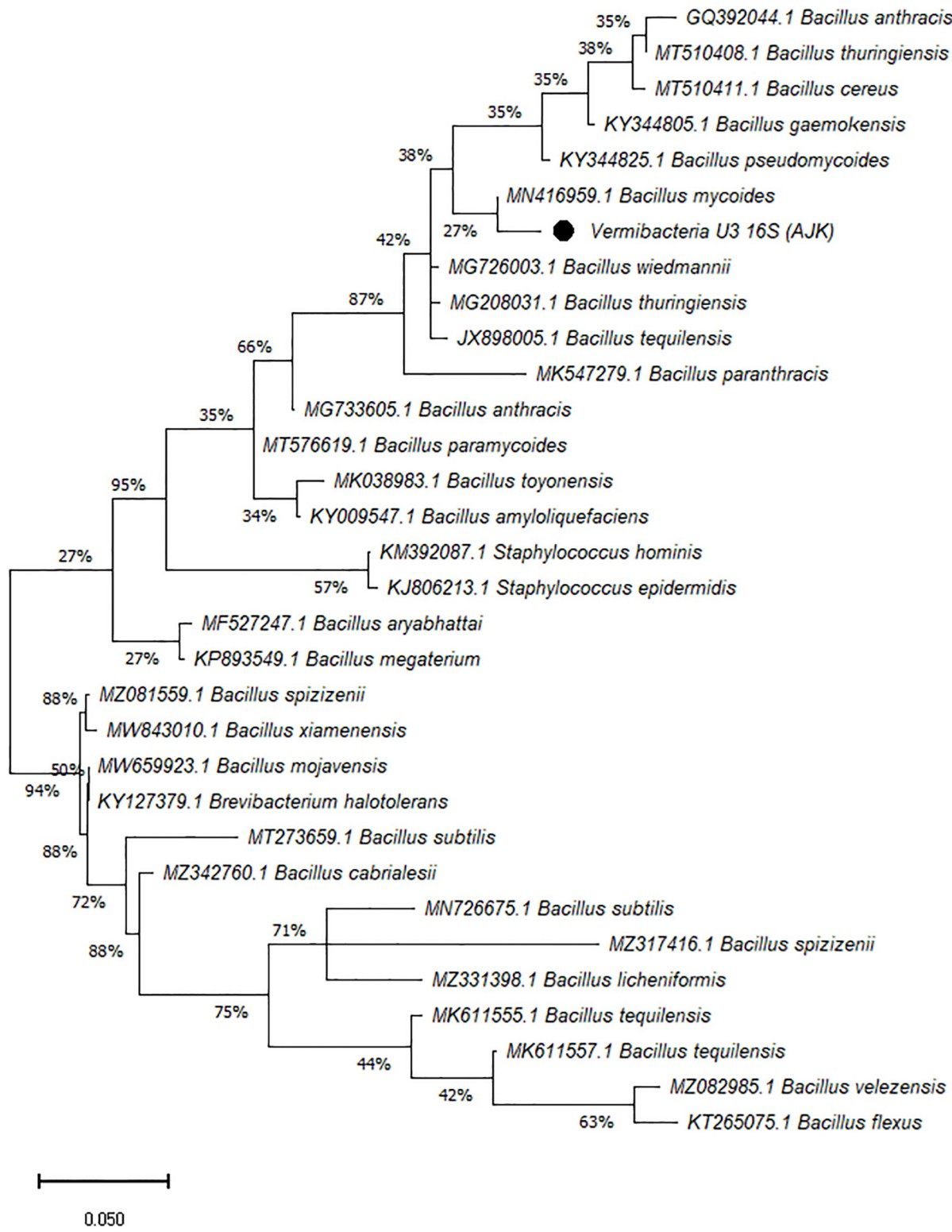

**Fig 11. Phylogenetic relationship of *Bacillus mycoides* with other known bacterial species.**

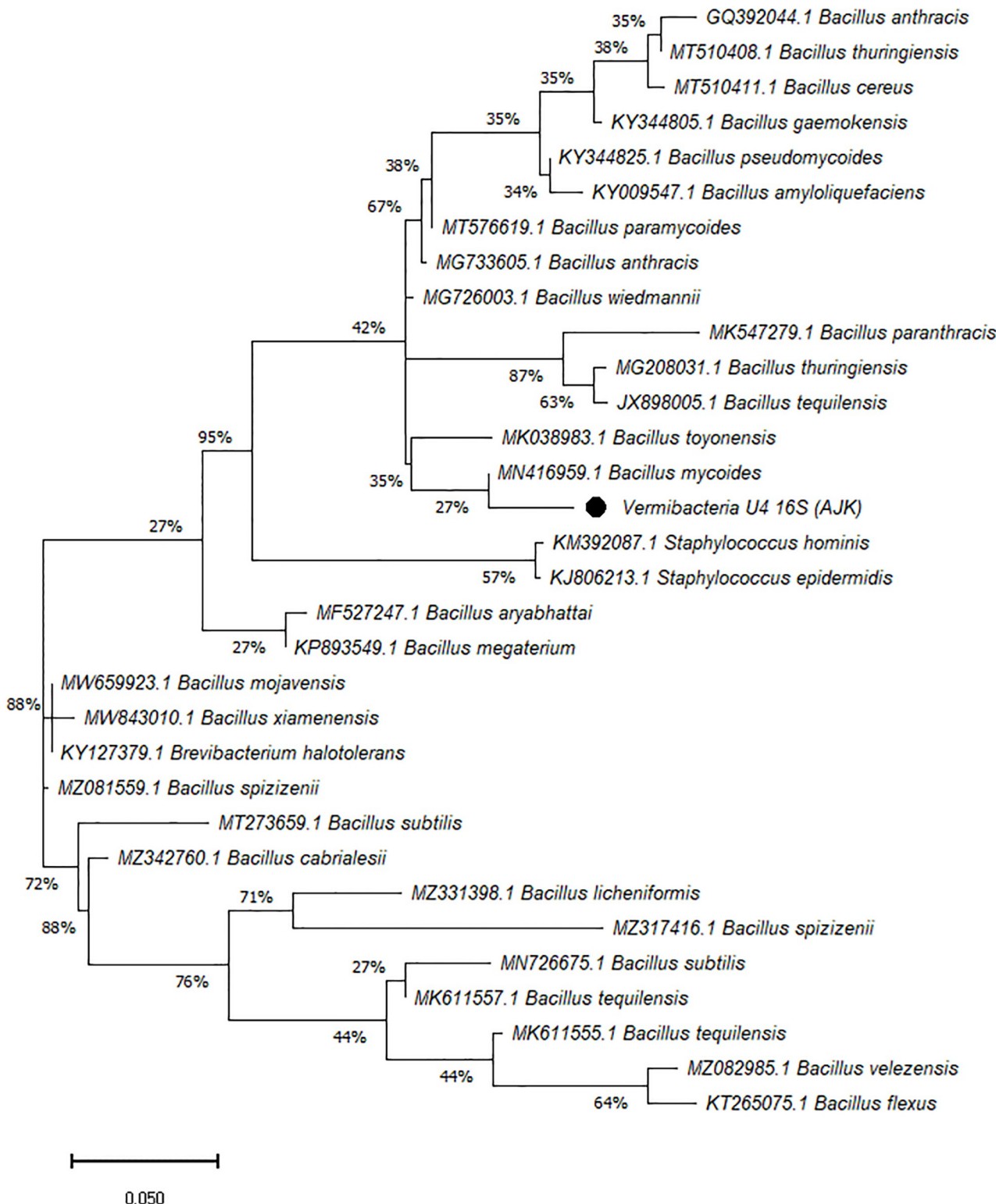

**Fig 12. Phylogenetic relationship of *Bacillus toyonensis* with other known bacterial species.**

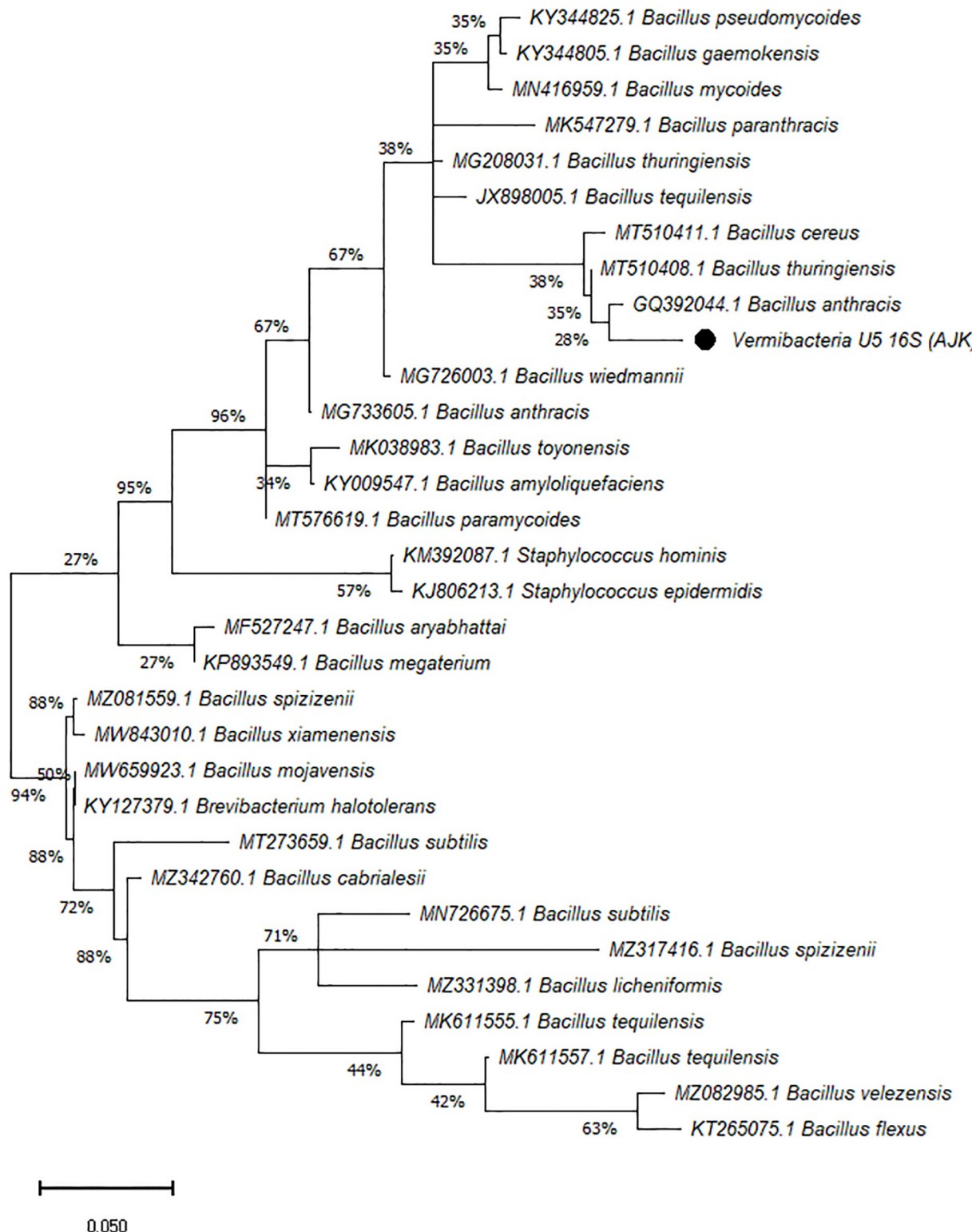

**Fig 13. Phylogenetic relationship of *Bacillus anthracis* with other known bacterial species.**

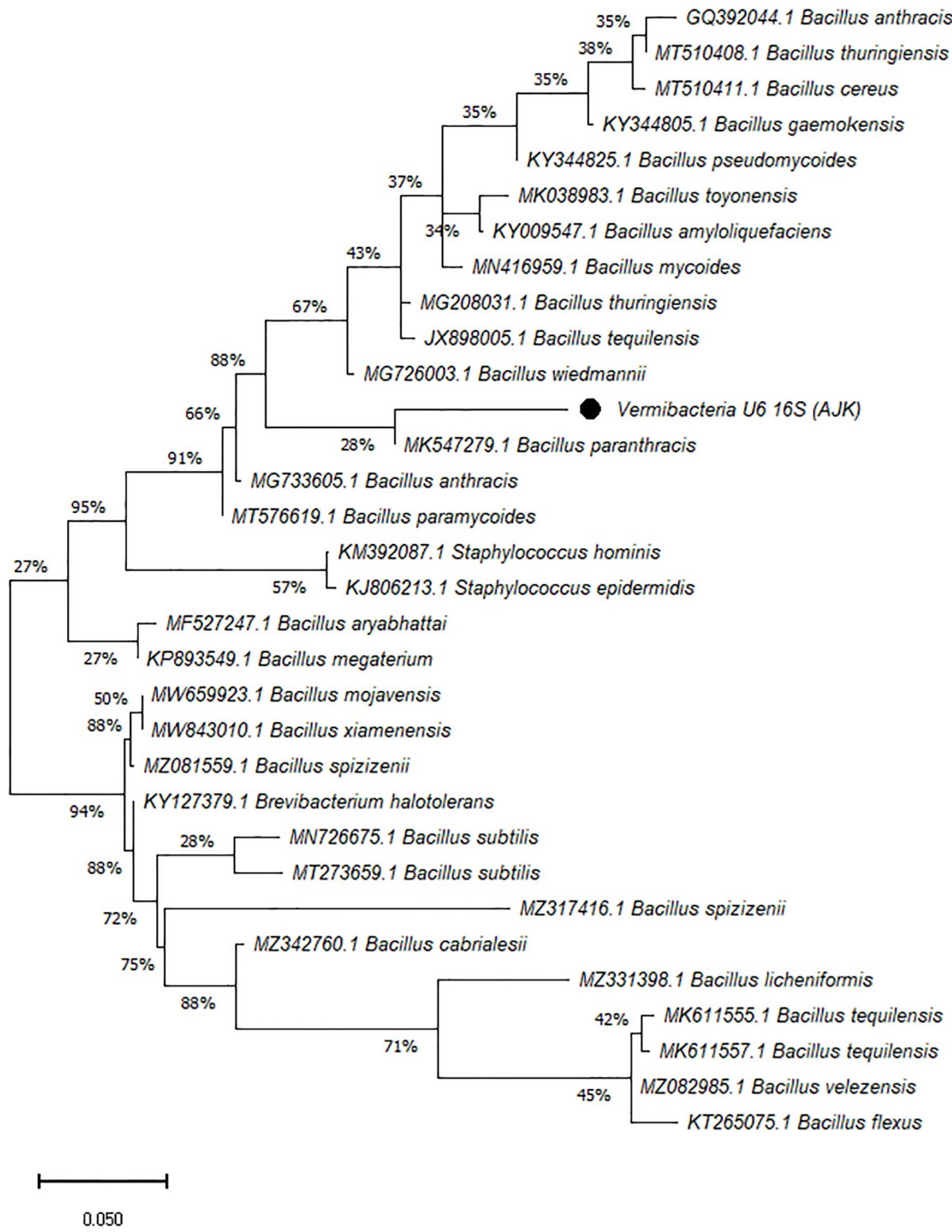

**Fig 14. Phylogenetic relationship of *Bacillus paranthracis* with other known bacterial species.**

species were coccobacilli and filamentous rods which was consistent with the results of Karsten and Drake, [41] who revealed the presence of cocci, rod-shaped bacteria, and filamentous microbes in *Lumbricus terrestris* L. and *Octolasion cyaneum* through scanning electron microscopy. Bacterial strains can be identified by growing them on selective culture media [42]. In current study, vermi-bacteria were grown on MacConkey agar and results revealed that *B. mojavensis*, *B. toyonensis*, *B. anthracis*, *S. hominis*, and *B. licheniformis* produced pink colonies indicating their ability to ferment lactose whereas *B. subtilis*, *B. paranthracis*, *B. mycoides* (B1, U3, and B4), *B. megaterium* are non-lactose fermenters. It was also observed that *B. subtilis*, *B. mojavensis*, *B. mycoides* have ability to ferment mannitol whereas *B. toyonensis*, *B. thruiginesis*, *B. paranthracis*, *B. mycoides* (B2 and B4), *B. megaterium*, *S. hominis*, and *B. licheniformis* showed no growth on mannitol salt agar. Our finding agreed with the outcomes of Silawat et al. [43]. They isolated and identified nine bacteria from the soil and compost. They showed that *P. aeruginosa*, *A. calcoacet*, *P. pseudomalli*, *P. pickettii*, *P. cepacia* have ability to ferment mannitol whereas *P. putida*, *P. shutzeri*, and *P. pickettii* fermented lactose.

Plant growth promoting bacteria (PGPB) play an essential role in the metabolism and growth of plants. Variety of PGPB such as *Bacillus*, *Arthobacter*, *Enterobacter*, *Azotobacter*, *Serratia*, and *Rhizobium* are being used worldwide to enhance the crop production [44]. Ammonia production is a significant trait of PGPR's and act as metabolic inhibitors towards phytopathogens [45, 46]. In the current research all vermi-bacteria have ability to produce ammonia except *S. hominis* and our findings are agreed with Kumar et al. [45] and could be used as antimicrobial agents. Our study revealed that all vermi-bacterial isolates are involved in IAA production except *B. mycoides* and *B. megaterium* suggesting the ability to suppress plant diseases, promote plant growth and development, and seed germination. Our results agreed with Khare and Arora, [47], who reported that bacterial indole acetic acid has a role in the suppression of rot diseases in various plants and IAA regulates several fundamental cellular processes including cell divisions, elongation, and differentiation. Our results are agreed with the previous literature [48–50]. They illustrated that IAA is produced by PGPB and helps in plant-microbe interactions.

Present work revealed that all vermi-bacterial isolates didn't produce HCN. Our findings are thus contrary to those of Nadège et al. [51] who isolated and identified nine PGPR i.e. five *Bacillus* species (*B. pantothenticus*, *B. circulans*, *B. thuringiensis*, *B. polymyxa* and *B. anthracis*) three *Pseudomonas* species (*P. cichorii*, *P. putida, and P. syringae*) and *Serratia marcescens* from the rhizospheric region of maize and all of this PGPR were capable of HCN production. The synthesis and production of HCN is varied from species to species and our findings agreed with the outcomes of Rijavec and Lapanje, [52]. Results revealed that all vermi-bacterial isolates produce siderophores and are phosphate solubilizers. Siderophores act as a biocontrol agent [53]. According to Indiragandhi et al. [54], siderophores form a stable complex with the trace elements (Fe, Ca, Zn, and Cu, etc.) which help in plant growth promotion. Similarly, phosphate solubilizing bacteria have a great impact on agriculture and are considered promising natural microbial biofertilizers [55]. *Bacillus megaterium* has been commercialized as BioPhos by AgriLife (India) [56]. Bhattacharyya and Jha, [57] also reported some phosphate solubilizing bacteria like *Beijerinckia*, *Erwinia*, *Azotobacter*, *Bacillus*, *Flavobacterium*, *Microbacterium*, *Burkholderia*, *Enterobacter*, *Rhizobium*, *Serratia*, and *Pseudomonas*, respectively.

In the current study, production of hydrolytic enzymes by PGPVB were screened and results revealed that all vermi-bacterial isolates were involved in the production of catalase, amylases, lipases, proteases, and oxidases that act as biocontrol agents to be used in the fields of medicine, environment and agriculture [58, 59]. These vermi-bacteria could be used in the plant disease management system. Our findings agreed with the outcomes of Parashar

et al. [60] who demonstrated that PGPB also produces antifungal agents. Protease and amylase-producing microbes such as *Pythium* spp and *Phytophthora* were not only played a major role in the plant growth promotion, decomposition of organic matter, and nutrient mineralization however also act as biocontrol agents [18]. Bacterial strains containing catalase action which showed resistance to environmental, chemical, and mechanical stress [45]. Our results revealed that all isolated vermi-bacteria were catalase-positive, and our results are parallel to the work of Silawat et al. [43] who reported that *P. aeruginosa*, *P. malli*, *Achromobacter group*, *P. putida*, *P. shutzeri*, *A. calcoacet*, *P. pseudomalli*, *P. pickettii*, *P. cepacia* isolated from compost were catalase positive. It was observed that vermi-bacterial isolates were oxidase-positive except *Bacillus mycoides (B1*, *B4 U3)*, *Bacillus/Priestia megaterium (B2)* and *Staphylococcus hominis* while our findings are contrary to the outcomes of Kaur and Brar, [61] who showed that *B. subtilis* is oxidase-negative. In our study, all bacteria were amylase positive except *Staphylococcus hominis*, parallel to the work of Geetha et al. [44] who confirmed that rhizospheric bacteria were able to produce amylases. The current research reveals that all isolated and identified plant growth promoting vermi-bacterial isolates showed the agricultural traits (siderophore, phytohormones, ammonia, and hydrolytic enzymes production, and also act as phosphate solubilizers). The current outcomes are consistent with the findings of previously reported data [56, 62–65].

## Conclusion

It was concluded that, earthworm gut is favorable host for the isolation of plant growth promoting bacteria which could not only be used as a microbial biofertilizers to enhance the crop production in Pakistan but also used in sustainable disease management system. Vermi-bacteria play an important role in certain soil processes such as growth hormone production, phosphorous solubilizers, nitrogen fixation, and control of microbial pathogens.

## Supporting information

**S1 Data.**
(DOCX)

## Acknowledgments

Authors are thankful to the Department of Biochemistry, Islamia University Bahawalpur, Bahawalpur for the molecular identification of vermi-bacterial isolates.

## Author Contributions

**Conceptualization:** Saiqa Andleeb.

**Data curation:** Saiqa Andleeb, Irsa Shafique, Anum Naseer, Wajid Arshad Abbasi, Samina Ejaz.

**Formal analysis:** Saiqa Andleeb, Irsa Shafique, Anum Naseer, Samina Ejaz, Iram Liaqat, Shaukat Ali, Muhammad Fiaz Khan, Fayaz Ahmed, Nazish Mazhar Ali.

**Funding acquisition:** Saiqa Andleeb.

**Investigation:** Saiqa Andleeb, Irsa Shafique, Wajid Arshad Abbasi, Muhammad Fiaz Khan.

**Methodology:** Saiqa Andleeb.

**Resources:** Iram Liaqat, Shaukat Ali, Nazish Mazhar Ali.

**Software:** Wajid Arshad Abbasi.

**Validation:** Shaukat Ali, Fayaz Ahmed.

**Writing – original draft:** Saiqa Andleeb.

**Writing – review & editing:** Irsa Shafique, Anum Naseer, Samina Ejaz, Shaukat Ali.

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
