## [Decision Letter · Decision Letter 0]

5 Apr 2022

PONE-D-21-33360Molecular Characterizatiion of plant growth promoting vermibacteria associated with Eisenia fetida gastrointestinal tractPLOS ONE

Dear Dr. Andleeb,

Thank you for submitting your manuscript to PLOS ONE. After careful consideration, we feel that it has merit but does not fully meet PLOS ONE’s publication criteria as it currently stands. Therefore, we invite you to submit a revised version of the manuscript that addresses the points raised during the review process.

We look forward to receiving your revised manuscript.

Kind regards,

Muhammad Hussnain Siddique

Academic Editor

PLOS ONE

https://journals.plos.org/plosone/s/file?id=ba62/PLOSOne_formatting_sample_title_authors_affiliations.pdf".

“Whole research work is done under HEC funded project No. TDF-02-006 titled: “Establishment of the vermi-tech unit at Azad Jammu and Kashmir, Muzaffarabad for vermi product development”.

“Whole research work is done under HEC funded project No. TDF-02-006 titled: “Establishment of the vermi-tech unit at Azad Jammu and Kashmir, Muzaffarabad for vermi product development”. The funders had no role in study design, data collection and analysis, decision to publish, or preparation of the manuscript.”

Reviewers' comments:

Reviewer's Responses to Questions

**Comments to the Author**

1. Is the manuscript technically sound, and do the data support the conclusions?

Reviewer #1: Yes

Reviewer #2: Yes

2. Has the statistical analysis been performed appropriately and rigorously? 

Reviewer #1: Yes

Reviewer #2: Yes

3. Have the authors made all data underlying the findings in their manuscript fully available?

Reviewer #1: Yes

Reviewer #2: Yes

4. Is the manuscript presented in an intelligible fashion and written in standard English?

Reviewer #1: Yes

Reviewer #2: No

5. Review Comments to the Author

Reviewer #1: 1- The literature review is adequate.

2- The experimental design is appropriate.

3- The manuscript with sound contribution.

4- The writing: It does contain many of the usual grammatical mistakes.

5- The discussion: Rewrite the last two sentences of the last paragraph. The conclusion was overstated beyond what the data actually support.

Reviewer #2: Reviewer Comments;

Title: Appropriate

Abstract:

Rephrase the sentence, “Results revealed that 11 vermi-bacterial isolates were isolated from the digestive tract of Eisenia fetida.”

Change “Indole acetic acid” to “indole acetic acid”

Rephrase the statement “It was concluded that all vermi-bacterial isolates could be used as potential microbial biofertilizers for the cultivation of crops in Pakistan.”

Abstract is poorly written, needs improvements

Introduction:

Provide latest citations/references

Material and Methods:

Ethical statement: ethical approval from institutional ethics committee is missing

Results:

Table 1: Scientific names must be on standard format “E. fetida”.

Discussion: Improve discussion, add latest citations

Conclusion: Conclusion is generalized. Provide specific outcomes of this study.

References: cross check references in the text and list provided

Language: improve manuscript for English language and grammatical errors

Comments for Editor:

Accept with major revision

6. PLOS authors have the option to publish the peer review history of their article (what does this mean?). If published, this will include your full peer review and any attached files.

Reviewer #1: No

Reviewer #2: No

---

## [Author Response · Author response to Decision Letter 0]

19 Apr 2022

Reviewer #1: 

1- The literature review is adequate. Thanks for positive consideration.

2- The experimental design is appropriate. Thanks for positive concern.

3- The manuscript with sound contribution. Thanks for positive comment

4- The writing: It does contain many of the usual grammatical mistakes. Has been modified

5- The discussion: Rewrite the last two sentences of the last paragraph. The conclusion was overstated beyond what the data actually support. Has been modified

Reviewer #2

Title: Appropriate

Thanks for positive consideration.

Abstract: 

Rephrase the sentence, “Results revealed that 11 vermi-bacterial isolates were isolated from the digestive tract of Eisenia fetida.”

Has been rephrased

Change “Indole acetic acid” to “indole acetic acid”

Has been changed

Rephrase the statement “It was concluded that all vermi-bacterial isolates could be used as potential microbial biofertilizers for the cultivation of crops in Pakistan.”

Has been rephrased

Abstract is poorly written, needs improvements 

Has been improved

Introduction:

Provide latest citations/references

Has been added.

Material and Methods:

Ethical statement: ethical approval from institutional ethics committee is missing 

Has been added

Results:

Table 1: Scientific names must be on standard format “E. fetida”. 

Has been modified

Discussion: Improve discussion, add latest citations 

Has been improved

Conclusion: Conclusion is generalized. Provide specific outcomes of this study. 

Has been modified

References: cross check references in the text and list provided

Has been modified

Language: improve manuscript for English language and grammatical errors 

Has been improved

---

## [Decision Letter · Decision Letter 1]

1 Jun 2022

Molecular Characterization of plant growth-promoting vermibacteria associated with Eisenia fetida gastrointestinal tract

PONE-D-21-33360R1

Dear Dr. Andleeb,

We’re pleased to inform you that your manuscript has been judged scientifically suitable for publication and will be formally accepted for publication once it meets all outstanding technical requirements.

Kind regards,

Muhammad Hussnain Siddique

Academic Editor

PLOS ONE

Additional Editor Comments (optional):

Reviewers' comments:

Reviewer's Responses to Questions

**Comments to the Author**

1. If the authors have adequately addressed your comments raised in a previous round of review and you feel that this manuscript is now acceptable for publication, you may indicate that here to bypass the “Comments to the Author” section, enter your conflict of interest statement in the “Confidential to Editor” section, and submit your "Accept" recommendation.

Reviewer #1: All comments have been addressed

Reviewer #2: All comments have been addressed

2. Is the manuscript technically sound, and do the data support the conclusions?

Reviewer #1: Yes

Reviewer #2: Yes

3. Has the statistical analysis been performed appropriately and rigorously? 

Reviewer #1: Yes

Reviewer #2: Yes

4. Have the authors made all data underlying the findings in their manuscript fully available?

Reviewer #1: Yes

Reviewer #2: Yes

5. Is the manuscript presented in an intelligible fashion and written in standard English?

Reviewer #1: Yes

Reviewer #2: Yes

6. Review Comments to the Author

Reviewer #1: The Introduction section requires English editing service. Improvement is required for Conclusion section.

Reviewer #2: (No Response)

7. PLOS authors have the option to publish the peer review history of their article (what does this mean?). If published, this will include your full peer review and any attached files.

Reviewer #1: **Yes: **Wei Hong Lau

Reviewer #2: No

---

## [Editor Report · Acceptance letter]

6 Jun 2022

PONE-D-21-33360R1 

Molecular Characterization of plant growth-promoting vermi-bacteria associated with *Eisenia fetida* gastrointestinal tract 

Dear Dr. Andleeb:

I'm pleased to inform you that your manuscript has been deemed suitable for publication in PLOS ONE. Congratulations! Your manuscript is now with our production department. 

Kind regards, 

on behalf of

Dr. Muhammad Hussnain Siddique 

Academic Editor

PLOS ONE